# Total Flavonoids in *Artemisia absinthium* L. and Evaluation of Its Anticancer Activity

**DOI:** 10.3390/ijms242216348

**Published:** 2023-11-15

**Authors:** Meizhu He, Kamarya Yasin, Shaoqi Yu, Jinyao Li, Lijie Xia

**Affiliations:** Xinjiang Key Laboratory of Biological Resources and Genetic Engineering, College of Life Science and Technology, Xinjiang University, Urumqi 830017, China; hmz1224@stu.xju.edu.cn (M.H.); kamarya@stu.xju.edu.cn (K.Y.); 107552201069@stu.xju.edu.cn (S.Y.)

**Keywords:** *Artemisia absinthium* L., total flavonoids, Cynaroside, Astragalin, ultrasonic-assisted extraction, enzymatic hydrolysis, antitumor activity

## Abstract

To overcome the shortcomings of traditional extraction methods, such as long extraction time and low efficiency, and considering the low content and high complexity of total flavonoids in *Artemisia absinthium* L., in this experiment, we adopted ultrasound-assisted enzymatic hydrolysis to improve the yield of total flavonoids, and combined this with molecular docking and network pharmacology to predict its core constituent targets, so as to evaluate its antitumor activity. The content of total flavonoids in *Artemisia absinthium* L. reached 3.80 ± 0.13%, and the main components included Astragalin, Cynaroside, Ononin, Rutin, Kaempferol-3-O-rutinoside, Diosmetin, Isorhamnetin, and Luteolin. Cynaroside and Astragalin exert their cervical cancer inhibitory functions by regulating several signaling proteins (e.g., EGFR, STAT3, CCND1, IGFIR, ESR1). Gene Ontology (GO) and Kyoto Encyclopedia of Genes and Genomes (KEGG) pathway enrichment analysis showed that the anticancer activity of both compounds was associated with the ErbB signaling pathway and FoxO signaling pathway. MTT results showed that total flavonoids of *Artemisia absinthium* L. and its active components (Cynaroside and Astragalin) significantly inhibited the growth of HeLa cells in a concentration-dependent manner with IC_50_ of 396.0 ± 54.2 μg/mL and 449.0 ± 54.8 μg/mL, respectively. Furthermore, its active components can mediate apoptosis by inducing the accumulation of ROS.

## 1. Introduction

Cervical cancer is one of the most common gynecologic malignancies worldwide and is a serious health risk for women due to its extremely high morbidity and mortality [1]. In 2020, 604,000 new cases and 342,000 deaths occurred worldwide [2]. The persistence of high-risk human papillomavirus (HPV) infection is a major factor in the development of cervical cancer [3]. Currently, the main treatment options for cervical cancer include surgical resection, radiation therapy, chemotherapy, locally targeted therapy, and immunotherapy [4,5,6]. Although conventional treatments are effective for early-stage cervical cancer, they have limited efficacy for locally staged and metastatic cervical cancer [7]. Most importantly, long-term use of chemotherapeutic agents, such as cisplatin, adriamycin, paclitaxel, and bleomycin, is prone to toxic side effects and drug resistance [8,9]. Therefore, biomedical researchers are also constantly searching for an alternative drug with better safety and effectiveness as a new strategy for cancer treatment.

Artemisia is a genus of small herbs and shrubs found in the northern temperate regions [10]. The species is often reported as a valuable additive for spice plants and the alcohol industry. In addition, the Artemisia species is rich in essential nutrients such as fatty acids, carbohydrates, dietary fiber, proteins, essential amino acids, vitamins, and minerals and thus is widely used as traditional food, tea, and dietary supplements [11,12]. Notably, the herbs of this species are also considered raw materials for oil extraction and have a high food value, as many of them are used in cooking [13,14]. Wei et al. [15] found that the extract of *Artemisia absinthium* inhibited the growth of hepatoma cells through the induction of apoptosis that might be mediated by endoplasmic reticulum stress and the mitochondria-dependent pathway and that three different organic-phase extracts effectively inhibited the growth of H22 cells in vivo in a mouse model of H22 tumors and improved the survival rate of tumor-bearing mice without significant toxicity. Meanwhile, Nazeri et al. [16] found that the methanolic extract of *Artemisia absinthium* triggered the apoptotic process in human colorectal cancer HCT-116 cells via downregulation of Bcl-2 and activation of caspase-3 and BAX. Shafi et al. [17] reported that *Artemisia absinthium* inhibits proliferation of human breast cancer MDA-MB-231 and MCF-7 cells through the induction of apoptosis by regulating Bcl-2 family proteins and MEK/MAPK signaling. Therefore, we chose this herb, *Artemisia absinthium* L., for the study of its antitumor activity.

*A. absinthium* L. belongs to Campanulales, family Asteraceae, and is commonly known as Wormwood [18]. This medicinal perennial is mainly distributed in Xinjiang, Sichuan, and Yunnan in China [19], and is rich in secondary metabolites including flavonoids, alkaloids, phenols, terpenoids, and derivatives with various pharmacological activities, such as antibacterial, anti-inflammatory, antiviral, hepatoprotective, antioxidant and anticancer [12,20,21,22,23,24]. In addition to its traditional medical applications, it has a high food value. Also, one of the most popular spirits of the 19th and 20th centuries, absinthe, whose main component is *A. absinthium* L., is a highly pure alcoholic beverage with psychoactive properties due to the high content of α- and β-tulip ketones [25,26].

Craciunescu et al. [27] showed that the total phenolic acid and total flavonoid contents of *Artemisia absinthium* L. extracts were significantly higher compared to *Arnica montana* L. and that in recent years, *Artemisia absinthium* L. has attracted great attention in cancer therapy due to its total phenolic and total flavonoid contents [16]. Total phenols and total flavonoids, which are important natural constituents of *Artemisia absinthium* L., showed significant inhibitory effects on the Huh-7 cell line but were not toxic to the Vero cell line [28]. Thus, we wanted to explore the antitumor mechanism of flavonoids in *Artemisia absinthium* L. Bioflavonoids, also known as flavonoids, are natural plant constituents [29]. More than 8000 species have been identified and isolated from plants, whose (C_6_-C_3_-C_6_) structures are formed through two phenolic hydroxyl groups of benzene rings (A and B rings), interconnected by a three-carbon bond [30,31]. They can be classified into six groups based on structural diversity, including flavones, isoflavones, flavanones, flavonols, flavanols, and anthocyanins [32]. Many studies have reported that flavonoids have a significant role in treating and preventing tumors [33,34,35,36]. The extraction methods of flavonoids mainly include heating, reflux extraction, soaking, organic solvent extraction, microwave-assisted extraction, ultrasound-assisted extraction, and enzymatic hydrolysis methods [37,38,39]. In the enzymatic hydrolysis method, cellulase is usually used to destroy the plant cell wall to release the intracellular material [40]. Ultrasonic extraction can continuously shake the extract to accelerate the dissolution of intracellular substances and improve the extraction rate of total flavonoids and the utilization rate of raw materials [41,42]. Although the traditional extraction method equipment is relatively simple, the extraction is time-consuming and inefficient. It is easy to destroy the thermally unstable components at high temperatures [43,44]. To overcome the shortcomings of the traditional extraction method and consider the low content and complexity of total flavonoids in *A. absinthium* L., in this experiment, we adopted ultrasound-assisted enzymatic hydrolysis to improve the yield of total flavonoids, which has the characteristics of mildness and environmental protection [45,46,47].

The response surface optimization method (RSM) has been widely used in process optimization based on a rational experimental design [48]. A multiple quadratic regression equation is used to fit the functional relationship between the factors and the response values. Finally, the regression equation is analyzed and predicted to obtain the best process parameters. Box–Behnken design (BBD) is a statistical model of the response surface design method. BBD represents an independent quadratic design that does not contain embedded or fractional factorial designs and is easy to design and statistically analyze compared to other designs [49,50]. It can reduce the number, cost, and time of experiments [51,52]. The main components include Astragalin, Cynaroside, Ononin, Rutin, Kaempferol-3-O-rutinoside, Diosmetin, Isorhamnetin, and Luteolin.

Network pharmacology is a systematic approach based on analyzing networks of disease, gene, protein targets, and drug interactions [53]. The synergistic effects of components and multi-target drugs and their potential mechanisms are better elucidated by analyzing complex and multi-level interaction networks. It has been widely used to study the molecular mechanisms of herbal medicines, herbal drug pairs, and herbal complexes acting on diseases [54]. Molecular docking is a method to design drugs by simulating the interaction pattern between receptors and drugs. In recent years, the application of molecular docking to explain the relevant mechanism of action has become a trend in developing new drugs [55]. In this study, we used a network pharmacology approach to explore the major bioactive components of total flavonoids in *A. absinthium* L. and predict their effective molecular targets and potential mechanisms in treating cervical cancer.

In summary, ultrasound-assisted enzymatic hydrolysis was used to optimize the extraction process of total flavonoids from *A. absinthium* L. based on single-factor experiments combined with the BBD experimental design, and multiple linear regression and binomial fitting of the experimental data were used to perform predictive analysis of the best process. The antitumor activity of the total flavonoids and their active ingredients (Cynaroside and Astragalin) obtained under the optimal extraction conditions were evaluated, showing good anti-cervical cancer activity. Furthermore, its active components can mediate apoptosis by inducing the accumulation of ROS. This study provided a theoretical basis for the further development and comprehensive application of *A. absinthium* L.

## 2. Results

### 2.1. Standard Curve

Based on the measurement results, the standard curve was plotted with the absorbance at 415 nm as the vertical coordinate and the quercetin concentration as the horizontal coordinate. The linear regression equation of total flavonoids in *A. absinthium* L. was obtained as y = 9.7675x − 0.0164, and the linear correlation coefficient R^2^ = 0.9974, suggesting that the standard had a good linear relationship.

### 2.2. Single-Factor Experiments

#### 2.2.1. Effect of Compound Enzyme Ratio on the Extraction Rate of Total Flavonoids

The fixed solid–liquid ratio was 1:10, the amount of the enzyme was 1.0%, pH was 4.0, the enzymatic hydrolysis temperature was 50 °C, the enzymatic hydrolysis time was 90 min, and the sonication time was 20 min. The effects of cellulase and pectinase at different ratios (1:3, 2:3, 1:1, 3:2, 3:1 g/g) on extracting total flavonoids in *A. absinthium* L. were investigated. As shown in Figure 1A, the total flavonoid yield showed a trend of increasing and then decreasing with the increased cellulase ratio. When the ratio of cellulase to pectinase was 3:2, the total flavonoid yield reached 2.68%. It is possible that the increased cellulase ratio may have a stronger effect on the damage of the cell wall and intercellular matrix of *A. absinthium* L., which effectively leads to the release of intracellular components. However, when the cellulase ratio reached a certain degree, some fibers were attached to the surface of *A. absinthium* L., which prevented the release of total flavonoids, resulting in a decrease in the total flavonoid yield. Therefore, the ratio of cellulose to pectinase was chosen to be 3:2.

#### 2.2.2. Effect of Enzyme Dosage on the Extraction Rate of Total Flavonoids

Based on the above-optimized compound enzyme ratio, the solid–liquid ratio was fixed at 1:10, pH was 4.0, the enzymatic hydrolysis temperature was 50 °C, the enzymatic hydrolysis time was 90 min and the sonication time was 20 min. The effects of the amount of compound enzyme (0.5%, 1.0%, 1.5%, 2.0%, 2.5%) on the total flavonoid yield of *A. absinthium* L. were investigated. As shown in Figure 1B, the total flavonoid yield significantly increased with the increase in the amount of the enzyme compound and then decreased. When the amount of enzyme was 2.0%, the total flavonoid yield reached the maximum value of 2.72%. Therefore, the optimal amount of the enzyme is 2.0%.

#### 2.2.3. Effect of pH on the Extraction Rate of Total Flavonoids

The optimum pH of cellulase for the experiment was between 4.0 and 6.5, and the optimum pH of pectinase was between 3.0 and 6.0. Based on the above-optimized ratio and amount of the complex enzymes, the solid–liquid ratio was fixed at 1:10, the enzymatic hydrolysis temperature was 50 °C, the enzymatic hydrolysis time was 90 min, and the sonication time was 20 min. The effects of pH (3.0, 3.5, 4.0, 4.5, 5.0) on the total flavonoid yield of *A. absinthium* L. were investigated. As shown in Figure 1C, the total flavonoid yield showed a trend of increasing and then decreasing with the increase in pH from 3.0 to 5.0. When the pH was 3.5, the total flavonoid yield reached a maximum of 3.07%. Therefore, a pH of 3.5 was chosen for the enzymatic hydrolysis.

#### 2.2.4. Effect of Ethanol Concentration on the Extraction Rate of Total Flavonoids

Based on the above-optimized ratio and dosage of complex enzymes and pH, the solid–liquid ratio was fixed at 1:10, enzymatic hydrolysis temperature was 50 °C, enzymatic hydrolysis time was 90 min, and sonication time was 20 min. The effects of ethanol concentrations (10%, 30%, 50%, 70%, 85%, and 100%) on the yield of total flavonoids in *A. absinthium* L. were investigated. As shown in Figure 1D, a lower ethanol concentration is generally suitable for extracting polar flavonoids, and a higher ethanol concentration is ideal for extracting non-polar flavonoids. When the concentration of ethanol increased from 10% to 85%, the extraction amount of total flavonoids was increased and reached the highest amount at 85% ethanol. Therefore, the optimum ethanol dosage was 85%.

#### 2.2.5. Effect of Solid–Liquid Ratio on the Extraction Rate of Total Flavonoids

Based on the above-optimized ratio and amount of complex enzymes, pH, and ethanol concentration, the enzymatic hydrolysis temperature was 50 °C, the enzymatic hydrolysis time was 90 min, and the sonication time was 20 min. The effects of solid–liquid ratios (1:5, 1:8, 1:10, 1:12, 1:15, 1:20 g/mL) on the total flavonoid yield of *A. absinthium* L. were investigated (Figure 1E). When the solid–liquid ratio was 1:10 (g/mL), it was difficult for the material to be completely immersed in the solvent, resulting in incomplete extraction. When the solid–liquid ratio was 1:15 (g/mL), the maximum yield of total flavonoids reached 1.24%. Therefore, 1:15 (g/mL) was chosen as the best solid–liquid ratio.

#### 2.2.6. Effect of Enzymatic Hydrolysis Temperature on the Extraction Rate of Total Flavonoids

Based on the above-optimized compound enzyme ratio and dosage, pH, ethanol concentration, and solid–liquid ratio, the enzymatic hydrolysis time was 90 min and the sonication time was 20 min. The effect of enzymatic hydrolysis temperature (30, 40, 50, 60, 70 °C) on the yield of total flavonoids in *A. absinthium* L. was investigated. As shown in Figure 1F, the total flavonoid yield showed a trend of increasing and then decreasing with the increase in enzymatic hydrolysis temperature. When the enzymatic hydrolysis temperature was 45 °C, the total flavonoid yield reached 3.51%. Therefore, the optimal temperature of enzymatic hydrolysis was 45 °C.

#### 2.2.7. Effect of Enzymatic Hydrolysis Time on the Extraction Rate of Total Flavonoids

Based on the above-optimized compound enzyme ratio and amount, pH, ethanol concentration, solid–liquid ratio, and enzymatic hydrolysis temperature, the effect of enzymatic hydrolysis time (45, 60, 75, 90, 105, 120 min) on the yield of total flavonoids in *A. absinthium* L. was investigated by fixing the sonication time of 20 min. The compound enzyme failed to destroy the cell wall sufficiently under a shorter enzymatic hydrolysis time, which resulted in a low total flavonoid yield. When the enzymatic hydrolysis time was extended to 105 min, the total flavonoid yield reached 4.59% (Figure 1G). Therefore, the enzymatic hydrolysis time of 105 min was chosen.

#### 2.2.8. Effect of Ultrasonic Temperature on the Extraction Rate of Total Flavonoids

Based on the above-optimized ratio and dosage of complex enzymes, pH, ethanol concentration, solid–liquid ratio, and enzymatic hydrolysis temperature and time, the effects of sonication temperature (20 °C, 40 °C, 60 °C, 80 °C, 100 °C, 120 °C) on the yield of total flavonoids in *A. absinthium* L. were investigated. The increase in sonication temperature has a certain promotional effect on the extraction of total flavonoids between 10 °C and 30 °C. The highest total flavonoid yield reached 4.75% at 30 °C (Figure 1H). Therefore, the optimal sonication temperature of 30 °C was selected.

### 2.3. Analysis of Extraction Parameters Using Response Surface Method

#### 2.3.1. Response Surface Experimental Design

Response surface methodology is an important method in condition optimization, and only a few experiments are required to obtain statistically acceptable results. Based on the single-factor experiments, three factors (solid–liquid ratio (A), enzymatic hydrolysis temperature (B), and ethanol concentration (C)) with significant effects on the extraction rate of total flavonoids from *A. absinthium* L. were selected for the three-level response surface method experiments (Table 1). The 17 response values for different combinations are shown in Table 2. Applying a quadratic linear regression fitted to the data and the significance and variance analysis yielded the following binary, multinomial regression model. Y = 3.81 − 0.19A + 3.424E − 003B + 0.16C − 0.27AB + 0.11AC − 0.026BC − 0.68A^2^ − 0.47B^2^ − 0.45C^2^. As shown in Table 3, the regression model was highly significant (*p* < 0.01), and the out-of-fit term *p* = 0.1379 was not significant (confidence level of 95%), indicating a good fit of the model. The correlation coefficient of the regression model equation R^2^ = 0.9373 indicates that the correlation of the predicted and true values is high, and the error is small, which can well reflect the experimental data. The CV value indicates the accuracy of the test, and the model’s CV value is 7.14%, suggesting the test is reliable. One equation A < 0.05 is significant, B and C > 0.05 is not significant, binomial A^2^, B^2^, and C^2^ < 0.01 is highly significant, and interaction term AB < 0.05 is significant, while AC and BC > 0.05 is not significant. It indicates that the effect of each experimental factor on the response value is not a simple linear relationship but a non-linear relationship. Examining the magnitude of the *p*-value and F-value, the degree of influence of each factor on the total flavonoid yield can be obtained as A > C > B.

#### 2.3.2. Response Surface Analysis

The steeper the response surface, the greater the effect of the factor on the total flavonoid yield and the more sensitive it is to the operating conditions. The contour lines can reflect the significance of the interaction between the factors. The oval shape indicates that the interaction between the two elements is significant, while the circle shows no significant change. As shown in Figure 2A, the response surface of the interaction term AB is steeper. The elliptical shape of the contour line shows that there is a significant interaction between the temperature of the enzymatic hydrolysis and the solid–liquid ratio that influences the rate of extraction of total flavonoids from *A. absinthium* L. Instead of increasing initially and then declining with an increase in enzymatic hydrolysis temperature, the extraction rate of total flavonoids increased progressively with the increase in solid–liquid ratio when the enzymatic hydrolysis temperature was fixed. The interaction terms AC (Figure 2B) and BC (Figure 2C) were relatively smooth, i.e., the interaction was weak, and the contours were approximately circular, indicating the interactions between the solid–liquid ratio and ethanol concentration and enzymatic hydrolysis temperature and ethanol concentration were not significant. The effect on the extraction rate of total flavonoids was not significant. The effect on the extraction rate of total flavonoids was negligible. The optimal conditions for extraction of total flavonoids from absinthe were solid–liquid ratio 1:14.6, ethanol concentration 87.45%, ultrasonic temperature 30 °C, enzyme ratio 3:2, enzyme dosage 2%, enzymatic hydrolysis temperature 45.19 °C, enzymatic hydrolysis time 105 min, and enzymatic hydrolysis pH 3.5. In order to facilitate the experimental operation, the ideal extraction conditions were established: solid–liquid ratio 1:15, ethanol concentration 85%, ultrasonic temperature 30 °C, enzyme ratio 3:2, enzyme dosage 2%, enzymatic hydrolysis temperature 45 °C, enzymatic hydrolysis time 105 min, enzymatic hydrolysis pH 3.5. The results showed that the yield of total flavonoid content was 3.8%. The results showed that the total flavonoid content of A can be extracted by the surface reaction method under optimum technical conditions.

### 2.4. LC-MS

LC-MS was used to qualitatively and quantitatively analyze the main components of total flavonoids in *A. absinthium*. The total ion flow chromatogram of total flavonoids showed that 41 major peaks were well separated (Figure 3). Based on the information of the excimer ion peaks given by the primary mass spectrometry and the structural fragmentation information of the secondary mass spectrometry, combined with the retention time of UPLC and the references, the possible structures of 41 chemical components were deduced, including 37 flavonoids, 2 phenolic compounds, 1 terpenoid and 1 amino acid and its derivatives (Figure 3B, Table 4).

### 2.5. Screening Mutual Targets of Drugs and Diseases

The core active ingredient targets of total flavonoids were matched with cervical cancer targets, from which the two main active ingredient compounds (Cynaroside and Astragalin) were selected to have six and five compound targets, respectively (Figure 4A,C). The target interaction PPI network graph (Cynaroside) was obtained with the STRING online analysis platform with 46 nodes and 331 edges, and the average node degree value was 14.4. The PPI network showed (Figure 4B) that EGFR, ESR, HSP90AA1, EP300, STAT3, EGF, and CCND1 were the main targets for cervical cancer. The PPI network graph (Figure 4D, Astragalin) had a total of 45 nodes and 279 edges with an average node degree value of 12.4. The PPI network showed that SRC, EGFR, ESR1, STAT3, EGF, CCND1, CDH1, and IGF1R were the main targets of cervical cancer. 

### 2.6. Enrichment Analysis of GO and KEGG Pathway

In the GO functional enrichment analysis of key targets, GO terms were screened according to *p* (*p* < 0.05, Q < 0.05) values. There were 590, 40, and 67 GO terms associated with biological processes, cellular components, and molecular functions (Cynaroside), respectively. As shown in Figure 5A, the first 20 biological processes were associated with the ERBB2 signaling pathway, epidermal growth factor, cellular response to estrogen stimulus, and mammary gland epithelial cell proliferation, among others. For cellular components, the targets are enriched in Chromatin and Nuclear chromatin. Molecular functional analysis revealed Type iii transforming growth factor beta receptor binding, Type i transforming growth factor beta receptor binding, Nitric-oxide synthase regulator activity, and Transcription regulator complex. Another active component had 418, 45, and 50 GO terms related to biological processes, cellular components, and molecular functions, respectively (Astragalin). As shown in Figure 5B, the first 20 biological processes related to the ERBB2 signaling pathway, entry of bacterium into host cell, regulation of epithelial cell proliferation involved in prostate gland development, epithelial cell differentiation involved in prostate gland development, and positive regulation of intracellular steroid hormone receptor signaling pathway. For cellular components, the targets are enriched in Flotillin complex, Fascia adherens, Beta-catenin destruction complex, beta-catenin-TCF complex, and Wnt signalosome. Molecular functional analyses revealed heat shock protein binding, Beta-catenin binding, Ephrin receptor binding, Hsp90 protein binding, and Tau protein binding. KEGG signaling pathway enrichment analysis contained 93 pathways (Cynaroside) and 83 pathways (Astragalin), respectively. Screening enrichment of the top 20 pathways included bladder cancer, pancreatic cancer, endocrine resistance, glioma, chronic myeloid leukemia, non-small cell lung cancer, prostate cancer, colorectal cancer, melanoma and FoxO signaling pathway (Figure 5C); and bladder cancer, endometrial cancer, prostate cancer, glioma, thyroid cancer, and ErbB signaling pathway (Figure 5D).

### 2.7. Active Ingredients–Shared Key Targets–Signal Pathway Network Diagram Construction

A component–target–channel network diagram was formed using Cytoscape 3.7.2 to comprehensively clarify the mechanism of active ingredients in cervical cancer (Figure 6). In the figure, the yellow diamond represents the component in the drug, the green oval represents the target of the component, and the pink V represents the pathway enriched in the target. It shows that each active compound can act on multiple targets, that each target can also respond to multiple active ingredients, and that each target can have an effect on multiple pathways.

### 2.8. Key Active Ingredients and Core Target Molecular Docking Results

Molecular docking results of key active ingredients and core targets. Molecular docking was performed for the core targets with the highest degree of PPI network and their corresponding active ingredients. The docked protein data were downloaded from the PDB database for EGFR (PDB ID 5wb7) and SRC (PDB ID 4hxj). The 3D structures were imported into AutoDock 1.5.7 software and docked with the active compounds. The core targets in the PPI network and their corresponding small molecule drug ligands were docked using AutoDock. The lower the binding energy, the better the ligand will bind to the protein. The molecules with the lowest binding energy were visualized using Pymol 2.2.0 software (Figure 7). The main form of interaction is hydrogen bonding. This result suggests that their combination may play an important role in the treatment of cervical cancer with total flavonoids.

### 2.9. Total Flavonoids and Active Ingredients Can Inhibit Cell Proliferation and Induce Apoptosis

To evaluate the antitumor effect of total flavonoids from *A. absinthium* L., HeLa cells were treated with different concentrations of total flavonoids and their active ingredients for 24 h. The morphological changes of the cells were observed by inverted microscopy. HeLa cells showed a small and round morphology, with a significant decrease in cell number. The active ingredients Cynaroside and Astragalin also showed similar results (Figure 8A). In addition, the inhibitory effect of total flavonoids on the proliferation of HeLa, SiHa, and H8 cells was examined, and the activity of Cynaroside and Astragalin was also examined against HeLa, respectively. As shown in Figure 8B, total flavonoids inhibited the growth of human cervical cancer cells HeLa and SiHa in a dose-dependent manner, and the 24 h semi-inhibitory concentration (IC_50_) values were 396.0 ± 54.2 μg/mL and 449.0 ± 54.8 μg/mL, respectively. Interestingly, total flavonoids had a weak effect on the growth of human immortalized human cervical normal epithelial cells H8. Meanwhile, Cynaroside and Astragalin also inhibited the activity of HeLa cells in a dose-dependent manner with IC_50_ of 263.4 ± 3.7 μg/mL and 282.3 ± 5.0 μg/mL (Figure 8B). The effect of total flavonoids on HeLa cell apoptosis was further detected. After Annexin V-FITC/PI staining, the assay was performed by flow cytometry. The total flavonoids significantly induced HeLa cell apoptosis in a dose-dependent manner while causing a low level of necrosis. The results suggested that total flavonoids of *A. absinthium* L. had good anti-cervical cancer activity. Subsequently, we further tested whether the inhibitory effect of Cynaroside and Astragalin on the growth of HeLa cells was also associated with the induction of apoptosis. The results showed that low and high concentrations inhibited the proliferation of HeLa cells. Cynaroside and Astragalin significantly increased the percentage of apoptotic HeLa cells in a concentration-dependent manner compared to the control group (Figure 8C). Apoptosis of HeLa cells was further observed by Hoechst 33342 staining after 24 h with Cynaroside and Astragalin. The results revealed that most untreated cells showed round, intact, and uniformly stained nuclei. In contrast, the drug-treated group showed a bright blue color with increasing concentration, as the nucleus structure was altered and the chromatin of the nucleus underwent concentration-dependent fixation and breakage (Figure 8D).

### 2.10. Induction of ROS Accumulation by Active Ingredients of Total Flavonoids

Reactive oxygen species (ROS), a signaling molecule in cell regulation, can induce apoptosis. ROS dose plays an important role not only in energy metabolism but also in cellular physiological processes. Therefore, DCF-DA, a specific fluorescent dye for ROS detection, was used as a fluorescent probe to assess the changes in intracellular ROS levels after Cynaroside and Astragalin treatment of HeLa cells. DCF-DA rapidly crosses the cell membrane and is converted to DCFH by intracellular lipases. After staining, Cynaroside and Astragalin-treated cells showed green fluorescence in a dose-dependent manner (Figure 9A). Flow cytometry results showed a dramatic increase in cell fluorescence after 24 h of treatment compared to the control group (Figure 9B). Together, these results suggest that ROS plays an essential role in Cynaroside and Astragalin-induced apoptosis in HeLa cells.

## 3. Discussion

To date, significant efforts have been made to develop antitumor drugs. Natural products have been given attention due to their considerable bioactivities and safety. The extraction methods of natural components mainly include heating, reflux, immersion, organic solvent extraction, microwave-assisted extraction, and enzymatic hydrolysis methods, etc. In order to improve the extraction rate of total flavonoids in *A. absinthium* L., the combination of the ultrasonic extraction method with the enzymatic hydrolysis method was performed in this study, which has been widely used in the extraction of active ingredients from plants due to advantages including convenience, efficiency, and environmental friendliness. Cellulase can decompose plant tissues under mild conditions and accelerate the release of active ingredients, and ultrasonics can accelerate the dissolution of intracellular substances. Therefore, the combination of the two methods greatly improves the extraction efficiency of plant active ingredients. To a certain extent, it also significantly reduces the extraction cost and is greener and healthier. In addition, we conducted the design test and data analysis of ultrasound-assisted enzymatic extraction of total flavonoids by a single-factor test based on response surface optimization. It overcomes the drawback that the orthogonal test design can only deal with discrete level values but cannot find the best combination of factors and the optimal response face value over the whole region. At the same time, it makes up for the lack of interaction effects and other mixed effects. It can also reduce the number of experiments and time consumption. Similarly, Saliha et al. used a response surface optimization ultrasound-assisted extraction method to extract total phenols from *A. absinthium* L. and determined the optimal conditions for maximum extraction of active ingredients [154]. Dai et al. [155] extracted flavonoids from *Saussurea* involucrate by ultrasonication, and Xu et al. [156] extracted total flavonoids from sea red fruits by enzymatic hydrolysis. Yun et al. [157] found that the extraction of flavonoids baicalein and wogonin from *Scutellaria baicalensis* roots by ultrasound-assisted enzymatic hydrolysis almost doubled the conversion rate and significantly increased the productivity compared to the enzymatic hydrolysis method. Cheng et al. [158] reported that the extraction rate of flavonoids by ultrasonic-assisted extraction could reach 2.980 mg/g. Ou et al. [159] found the highest total flavonoids content achieved was 1.99 ± 0.02 g RE/100 g by a combination of supercritical CO_2_ and ultrasound. Thus, we effectively extracted the total flavonoids from *A. absinthium* L. by ultrasound-assisted enzymatic digestion in a time-saving manner.

Furthermore, total flavonoids of *A. absinthium* L. were prepared using the optimal extraction conditions, and their antitumor activity was evaluated. The results showed that total flavonoids of *A. absinthium* L. had good anti-cervical cancer activity. Sultan et al. found that extracts of *A. absinthium* L. could promote apoptosis in colon cancer cells through activation of the mitochondrial pathway [16]. It was also reported that extracts of *A. absinthium* L. were able to mediate the growth and apoptosis of hepatocellular carcinoma cells through endoplasmic reticulum stress and mitochondria-dependent pathways [9]. In addition, extracts of *A. absinthium* L. have anticancer and antioxidant effects on various tumors by increasing the intracellular amount of free radicals in cancer cells, which leads to DNA damage and, consequently, apoptosis [23]. As such, the anticancer activity of the active components of total flavonoids was also evaluated. In previous studies, it was confirmed that apoptosis is a physiological process of cells. Cancer occurs when this balance is disrupted, either by an increase in cell proliferation or a decrease in cell death [160]. Moreover, apoptosis induction is considered the primary pathway by which biological agents inhibit the proliferation of tumor cells. Thus, our study tested whether the inhibition of the proliferation of cervical cancer HeLa cells by Cynaroside and Astragalin was achieved through apoptosis. As we expected, Cynaroside and Astragalin significantly increased the rate of apoptosis in HeLa cells in a dose-dependent manner. The results suggest that Cynaroside and Astragalin inhibit the proliferation of cervical cancer cells by inducing apoptosis. In addition, we found that Cynaroside and Astragalin apoptosis induction may be associated with intracellular ROS levels in HeLa cells, and the excessive production of ROS leads directly to apoptosis. Therefore, ROS accumulation is considered one of the main targets of cancer therapy. Yuan et al. [161] found that Isoliquiritigenin induced apoptosis by increasing intracellular ROS levels in HeLa cells. Lim et al. [162] found that indole-3-carbinol mediated apoptosis in lung cancer H1299 cells by increasing ROS production. These are consistent with our results, which showed that Cynaroside and Astragalin inhibit the development and progression of cervical cancer HeLa cells by increasing the accumulation of ROS.

## 4. Materials and Methods

### 4.1. Materials and Reagents

*A. absinthium* L. powder (Purchased from Xinjiang Urumqi Tianyi Feng Biotechnology Co., Ltd., Urumqi, China, provided by Xinjiang Baokang Pharmaceutical Co., Ltd., and crushed to 200 mesh by high-speed universal pulverizer). The source of the *A. absinthium* L. provided by Xinjiang Baokang Pharmaceutical Co., Ltd. is in line with the requirements of the Plant Protection Law of the People’s Republic of China and the Biosafety Law of the People’s Republic of China and other relevant regulations and standards: the quality of the product is in line with the Drug Administration Law, the Good Manufacturing Practices for Drugs, the Good Manufacturing Practice for Drugs, and the Good Manufacturing Practices for Drugs, and other relevant regulations and standards. Ethanol, quercetin control (Shanghai Maclean Biochemical Technology Co., Ltd., Shanghai, China), sodium nitrite, aluminum nitrate, sodium hydroxide, aluminum trichloride, potassium acetate, methanol, ethyl acetate, petroleum ether, isopropanol (Tianjin Xinbote Chemical Co., Ltd., Tianjin, China).

### 4.2. Methods

#### 4.2.1. Quercetin Standard Curve Plotting

Quercetin 5.0 mg was weighed precisely and fixed to 25 mL with methanol to obtain a mass concentration of 0.2 mg/mL standard solution, which was stored in a refrigerator at 4 °C for use. Amounts of 0.1 mL, 0.2 mL, 0.3 mL, 0.4 mL, and 0.5 mL of quercetin control solution were added into 10 mL volumetric flasks, and 0.1 mol/mL AlCl_3_ solution was added for 6 min, then 1 mol/mL C_2_H_3_KO_2_ solution was added for 3 mL, and finally the volume with methanol solution was fixed to the scale. The corresponding solution without quercetin was used as a blank control, 100 μL was aspirated, and the absorbance at 415 nm was measured. The standard curve was plotted with absorbance (y) as the vertical coordinate and quercetin concentration (x) as the horizontal coordinate.

#### 4.2.2. Extraction of Total Flavonoids

After weighing 2.0 g of *A. absinthium* L. powder in a 50 mL centrifuge tube, ethanol solution at a ratio of 1:10 was added, adding a certain amount of complex enzymes (cellulase and pectinase) according to the mass ratio of *A. absinthium* L., carrying out enzymatic hydrolysis in a water bath at a certain pH and temperature (mixed once at 30 min), and then inactivating the enzyme in a water bath at 100 °C for 5 min, adding 10 mL of ethanol solution at a concentration of 80%, and finally sonicating the above enzymatic solution at a certain power, sonication temperature, and sonication time. Then, 10 mL of ethanol solution was added, and the above enzymatic solution was sonicated at a certain ultrasonic power, ultrasonic temperature, and ultrasonic time. Finally, the supernatant was taken from the sonicated sample solution after centrifugation, and the absorbance of each solvent was measured at 415 nm with a blank as the control, and the total flavonoid extraction rate and antitumor activity were calculated. (The blank was divided into those with and without enzyme, and the effect of enzyme addition on the yield and activity was evaluated before and after the enzyme addition).

#### 4.2.3. Determination of Total Flavonoid Content

Weigh accurately 2.0 g of the dried powder of *A. absinthium* L. crushed with a high-speed universal grinder, add a certain volume fraction of aqueous ethanol solution according to a certain solid–liquid ratio; extract with ultrasound for a certain time in an ultrasonic cleaner, filter, and concentrate the extract with a rotary evaporator under reduced pressure to remove the solvent. The concentrated solution was transferred to a 100 mL volumetric flask, and the volume was fixed with ethanol to the scale. Precisely measure 1.0 mL of *A. absinthium* L. flavonoids extract with aluminum trichloride-potassium acetate method to detect the absorbance value (A) of total flavonoid extract and bring to the standard. The curve regression equation calculated the total flavonoid concentration and then converted it into flavonoid content.

Total flavonoid extraction amount (mg/g) = C × V × N/M

C: concentration of total flavonoids measured (mg/mL);

V: volume of extraction solution (mL);

N: dilution times;

M: mass of raw material (g).

#### 4.2.4. Single-Factor Experimental Design

Weighing 2.0 g of *A. absinthium* L., the factors that affected the total flavonoid extraction were selected by the controlled variable method for the one-way experiment. Some of the factors are ethanol concentration, enzyme complex ratio, enzyme complex amount, pH, material-liquid ratio, enzyme hydrolysis temperature, enzyme hydrolysis time, sonication temperature, and sonication time. The influence of other factors on the extraction rate of flavonoids from *A. absinthium* L. was investigated using a single-factor experimental design.

### 4.3. Optimization of Extraction Conditions of Total Flavonoids by Response Surface Methodology

The results of the single-factor experiments were combined. Three factors, including solid–liquid ratio, enzymatic hydrolysis temperature, and ethanol content, were selected to significantly affect the extraction rate of total flavonoids from *A. absinthium* L. Regression analysis was performed using DesignExpert 8.0 software to calculate the optimal extraction conditions using a regression equation, and validation test was performed three times in parallel to obtain the optimal extraction conditions by the response surface method and complete analysis of each factor.

### 4.4. LC-MS

#### 4.4.1. Metabolites Extraction

The sample was thawed on ice. After 30 s vortex, the 200 μL sample was dried under gentle nitrogen flow. The sample was then reconstituted in 200 μL of 50% methanol containing 0.1% formic acid. After vortexing for 30 s, it was subjected to ultrasound in an ice bath for 15 min. The sample was then centrifuged at 12,000 rpm (RCF = 13,800× *g*, R = 8.6 cm) for 15 min at 4 °C. The resulting supernatant was passed through a 0.22 μm filter membrane. It was then transferred to 2 mL glass vials and stored at −80 °C until analysis by UHPLC-MS/MS.

#### 4.4.2. MS Analysis

AB Sciex QTrap 6500+ mass spectrometer was applied for assay development. Typical ion source parameters were IonSpray Voltage: +5000/−4500 V, Curtain Gas: 35 psi, Temperature: 500 °C, Ion Source Gas 1: 55 psi, Ion Source Gas 2: 60 psi.

#### 4.4.3. Screening of Core Active Ingredients and Targets

PharmMapper (http://lilab-ecust.cn/pharmmapper/, accessed on 6 April 2022) and SwissTargetPrediction databases (http://www.swisstargetprediction.ch/, accessed on 6 April 2022) were used to retrieve the gene targets of the active ingredients. The obtained targets were then mapped to the UniProt database (https://www.uniprot.org/, accessed on 6 April 2022) for normalization.

#### 4.4.4. Disease-Related Gene Mining

Targets associated with cervical cancer were collected from the OMIM database (https://www.omim.org/, accessed on 6 April 2022) and DrugBank database (https://www.drugbank.ca/, accessed on 6 April 2022). Normalization of gene names and definition of species as “human” was performed using the UniProt database. Venn diagrams of active ingredients and cervical cancer targets were mapped using a bioinformatics platform.

#### 4.4.5. Protein–Protein Interaction Network Construction and Analysis

A target PPI network was obtained by compiling potential targets from the STRING database (https://string-db.org/, accessed on 6 April 2022). The species selection parameter was set to “Homo sapiens”, and the minimum required confidence in the interaction score was set to 0.900. The PPI network was then visualized using Cytoscape 3.7.2 software.

#### 4.4.6. Gene Ontology (GO) and the Kyoto Encyclopedia of Genes and Genomes (KEGG) Enrichment

Gene Ontology (GO) and Kyoto Encyclopedia of Genes and Genomes (KEGG) enrichments were performed using Bioconductor Cluster Profiler for core drug group ontology (GO) enrichment analysis of potential target genes and KEGG pathway enrichment analysis of tissues. Pathways were ranked according to the number of molecules in the pathway with a critical value (*p* < 0.05).

#### 4.4.7. Network Construction

Compound target pathway networks were constructed using Cytoscape 3.7.2. Topological properties were analyzed using Cytoscape’s Network Analyzer plug-in to identify key components and targets.

#### 4.4.8. Molecular Docking Ensures the Interaction between Targets and Key Compounds

Molecular docking ensures interactions between targets and key compounds. Target protein structures were obtained from the RCSB PDB database (http://www.rcsb.org/, accessed on 6 April 2022). Two-dimensional structures of ginseng leaf actives were sought from the PubChem database. ChemBio Draw 3D and Autodock Tool were applied to optimize the structures of key compounds and targets, including 3D chemical structure creation, energy minimization, and format conversion. PyMol was used to process proteins, including removal of ligands, correction of protein structures, and removal of water. Docking was carried out using R 4.1.3 software and Autodock Vina to select the molecule with the lowest binding energy in the docked conformation and observe binding effects by matching to the original ligand and intermolecular interactions.

### 4.5. Cell Culture

HeLa, SiHa, and H8 cells were obtained from the Xinjiang Key Laboratory of Biological Resources and Genetic Engineering, Xinjiang University (Urumqi, Xinjiang, China) and were cultured in RPMI1640 with DMEM (Gibco, Thermo Fisher Scientific, Waltham, MA, USA) medium having 10% fetal bovine serum (MRC, Changzhou, China), 50 U/mL each of penicillin and streptomycin (MRC, Changzhou, China), respectively, at 37 °C in a humidified environment with 5% CO_2_. 

### 4.6. MTT

HeLa and SiHa cells at the logarithmic stage were inoculated with 5 × 10^3^ cells /mL in 96-well plates and treated with different concentrations of total flavonoids (100, 200, 400, 500, 600, and 800 μg/mL) and Cynaroside and Astragalin (100, 150, 200, 250, 300, 350, and 400 μg/mL). Control (blank), 0.3%DMSO (negative control), and 35 μg/mL cisplatin (positive control) were set, with 6 replicates in each group. After 24 h of incubation, the waste liquid was discarded by centrifugation, and 100 μL MTT (0.5 mg/mL) was added to each of them and incubated at 37 °C for 4 h under dark conditions. The supernatant was removed by centrifugation, dissolved with 150 μL DMSO, and the OD value of each well was detected at 490 nm. The following formula calculated cell survival rate: cell viability (%) = (OD_treated_/OD_control_) × 100%.

### 4.7. Apoptosis

HeLa cells were inoculated with 1 × 10^5^ cells/dish in 60 mm culture dishes, and the cells were treated with different concentrations of total flavonoids (200, 400, and 600 μg/mL), Cynaroside, and Astragalin (200, 300, and 400 μg/mL) for 24 h, respectively. The digested cells were collected from each group, centrifuged at 1200 r/min for 7 min, washed twice with PBS (4–5 mL), and the supernatant was discarded. An amount of 100 μL of binding buffer (1 × binding buffer) was added to each group to resuspend the cells. Cells were resuspended by adding 2.5 μL Annenxin V-FITC and 5 μL PI, mixed and shaken for 10 min in an ice bath, resuspended by 300 μL of 1 × binding buffer, and detected by FASC. The data were analyzed using FlowJo7.6 software.

### 4.8. Hochest

HeLa cells were inoculated 1 × 10^5^ cells/well in 6-well plates and incubated for 24 h. Cells were treated with different concentrations of flavonoids, kynaroside, and astragalin and incubated at 37 °C for 24 h. The supernatant was discarded and washed twice with precooled PBS buffer, followed by fixation with precooled 4% paraformaldehyde for 10 min at 4 °C. After washing with PBS buffer, the cells were stained with Hoechst 33258 dye for 20 min in the dark. After washing with PBS, the cells were stained with Hoechst 33258 for 20 min under low light. After dyeing for 20 min, the cells were slowly washed twice with PBS for 3 min each time and photographed with an inverted fluorescent microscope to observe the changes in the karyotype.

### 4.9. ROS

Logarithmic growth stage HeLa cells were inoculated at a density of 1 × 10^5^/mL in 60 mm dishes for 24 h for cell apposition. HeLa cells were treated with various concentrations of total flavonoids, Cynaroside, and Astragalin and incubated at 37 °C for 24 h. Cells were collected, and the supernatant was discarded. Cells were suspended in the DCFH-DA dilution solution and incubated for 20 min at 37 °C. Cells were shaken every 3–5 min. The cells were washed 3 times with serum-free medium to adequately remove the unbound DCFH-DA fluorescent probe, which was detected by flow cytometry and analyzed by FlowJo 7.6 software.

### 4.10. Statistics

The experimental results were processed using Graphpad Prism 8 software, and the Box–Behnken experimental design and data analysis were performed using DesignExpert 8.0 software. All data were expressed as the mean ± standard deviation (S.D.). *p* < 0.05 was considered statistically significant.

## 5. Conclusions

In summary, *Artemisia* L. is traditionally used in food, spices, condiments, and beverages. Its leaves are the most commonly used edible part, and the above-ground part known as a “herb” is also widely used. Thus, we have extracted and optimized the extract of *A. absinthium* L. The optimal extraction conditions of the process were as follows: enzyme ratio 3:2, enzyme dosage 2%, enzymatic hydrolysis temperature 45 °C, enzymatic hydrolysis time 105 min, pH 3.5, solid–liquid ratio 1:15, ethanol volume percentage 85%, and sonication temperature 30 °C. The actual extraction rate of total flavonoids under these conditions could reach 3.80 ± 0.13%, which was similar to the total flavonoid extraction predicted by the model (3.84%). The optimized extraction process was simple, stable, and feasible to provide a theoretical basis for the further development and comprehensive application of *A. absinthium* L. Moreover, results from network pharmacology and molecular docking indicate that EGFR and SRC are the key targets of the two core components of total flavonoids against cervical cancer, so the total flavonoids of *A. absinthium* L. might be used to treat cervical cancer. In addition, both Cynaroside and Astragalin inhibited the proliferation of HeLa cells and induced apoptosis by increasing the production of ROS.

## Figures and Tables

**Figure 1 ijms-24-16348-f001:**
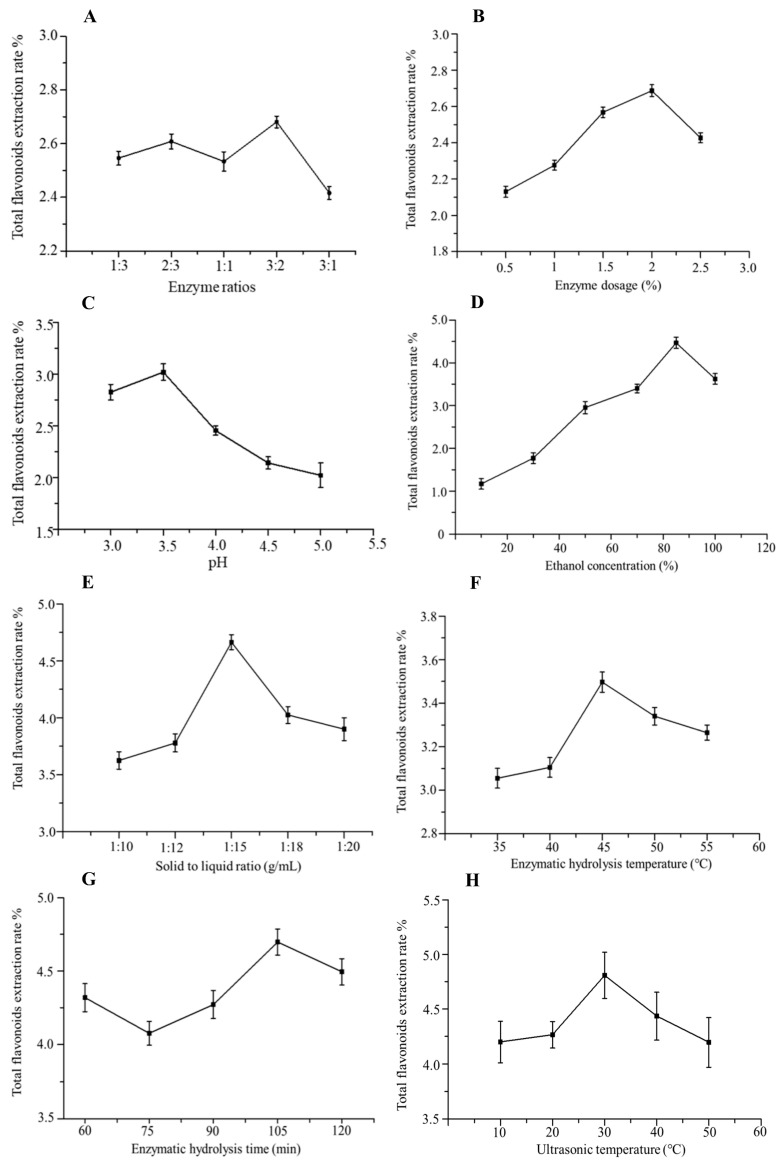
Effect of single factors on the extraction of total flavonoids from *A. absinthium* L. (**A**) Enzyme ratio; (**B**) enzyme dosage; (**C**) pH; (**D**) ethanol concentration; (**E**) solid–liquid ratio; (**F**) enzymatic hydrolysis temperature; (**G**) enzymatic hydrolysis time; (**H**) ultrasonic temperature.

**Figure 2 ijms-24-16348-f002:**
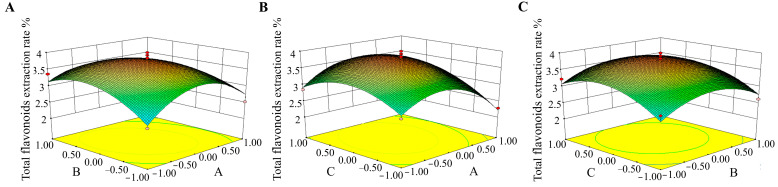
Response surface plots (3D) show different extraction parameters’ effects on the total phenolic yield. (**A**) Solid–liquid ratio and enzymatic hydrolysis temperature; (**B**) solid-to-liquid ratio and ethanol concentration; (**C**) enzymatic hydrolysis temperature and volume fraction of ethanol.

**Figure 3 ijms-24-16348-f003:**
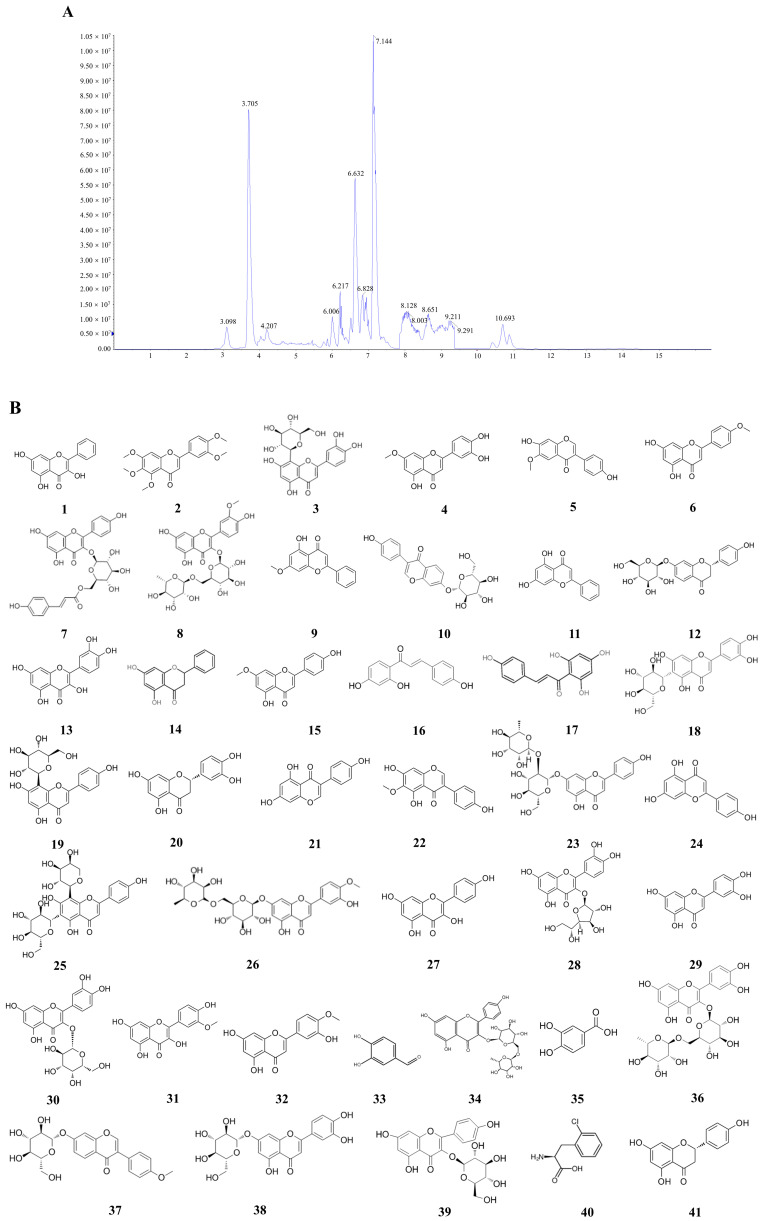
(**A**) Ion flow chromatogram of total flavonoid extract from *A. absinthium* L. (**B**) Structural formulae of compounds in the total flavonoid extract of *A. absinthium* L.

**Figure 4 ijms-24-16348-f004:**
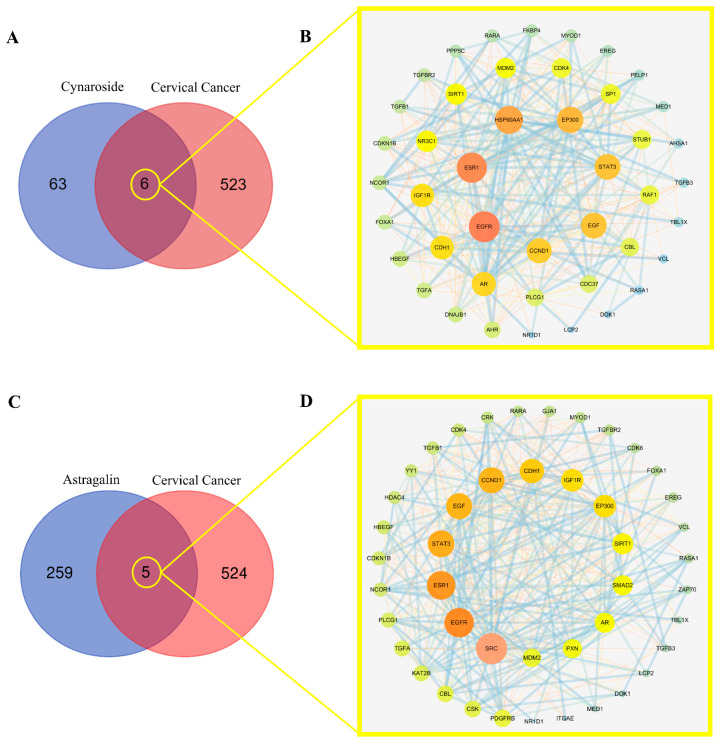
(**A**) Venn diagram of the total flavonoid component Cynaroside and cervical cancer target genes. (**C**) Venn diagram of the total flavonoid component Astragalin and cervical cancer target genes. (**B**,**D**) Shared Target Interactive PPI Network.

**Figure 5 ijms-24-16348-f005:**
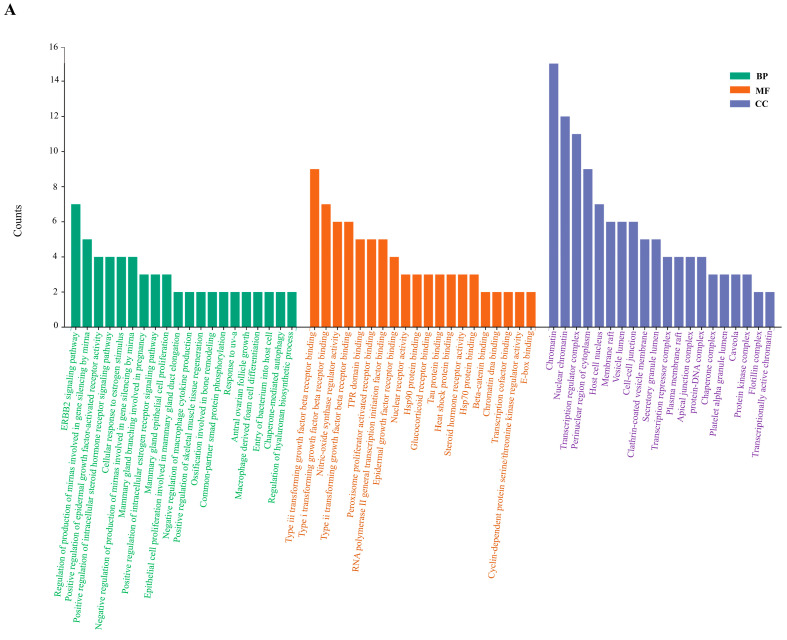
(**A**,**B**) The GO enrichment analysis of core nodes. Including cellular components, molecular functions, and biological processes. (**C**) The top 20 pathways for KEGG enrichment analysis of the targets of Cynaroside. (**D**) The top 20 pathways for KEGG enrichment analysis of the targets of Astragalin.

**Figure 6 ijms-24-16348-f006:**
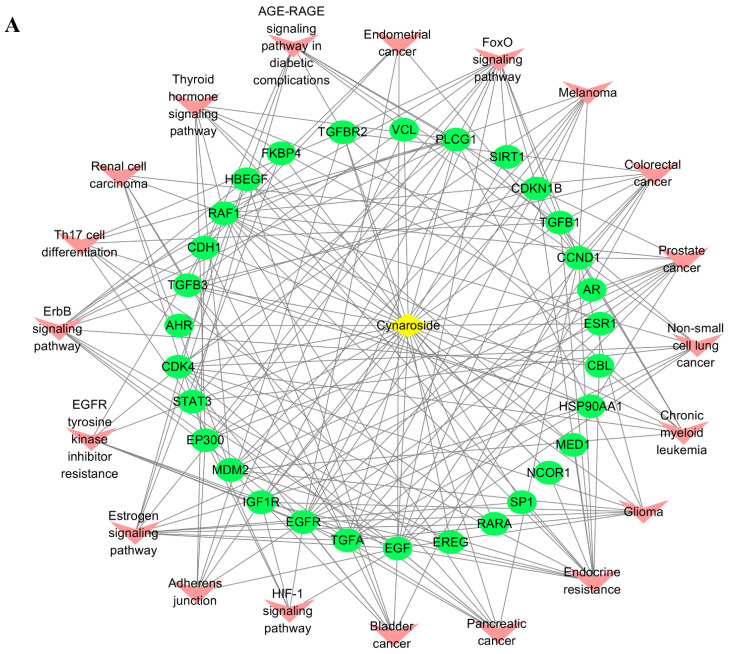
(**A**) Total Flavonoid Component Cynaroside–Target–Disease Network Diagram. (**B**) Total Flavonoid Component Astragalin–Target–Disease Network Diagram.

**Figure 7 ijms-24-16348-f007:**
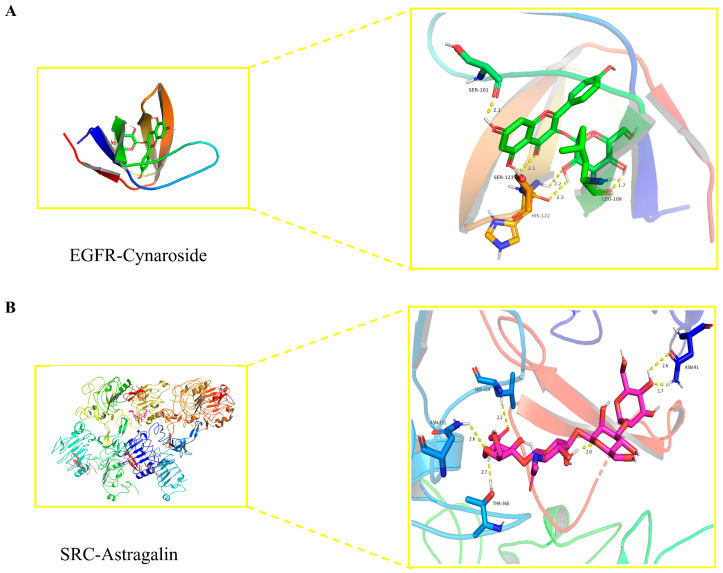
Detailed target–compound interactions. (**A**) EGFR–Cynaroside. (**B**) SRC–Astragalin.

**Figure 8 ijms-24-16348-f008:**
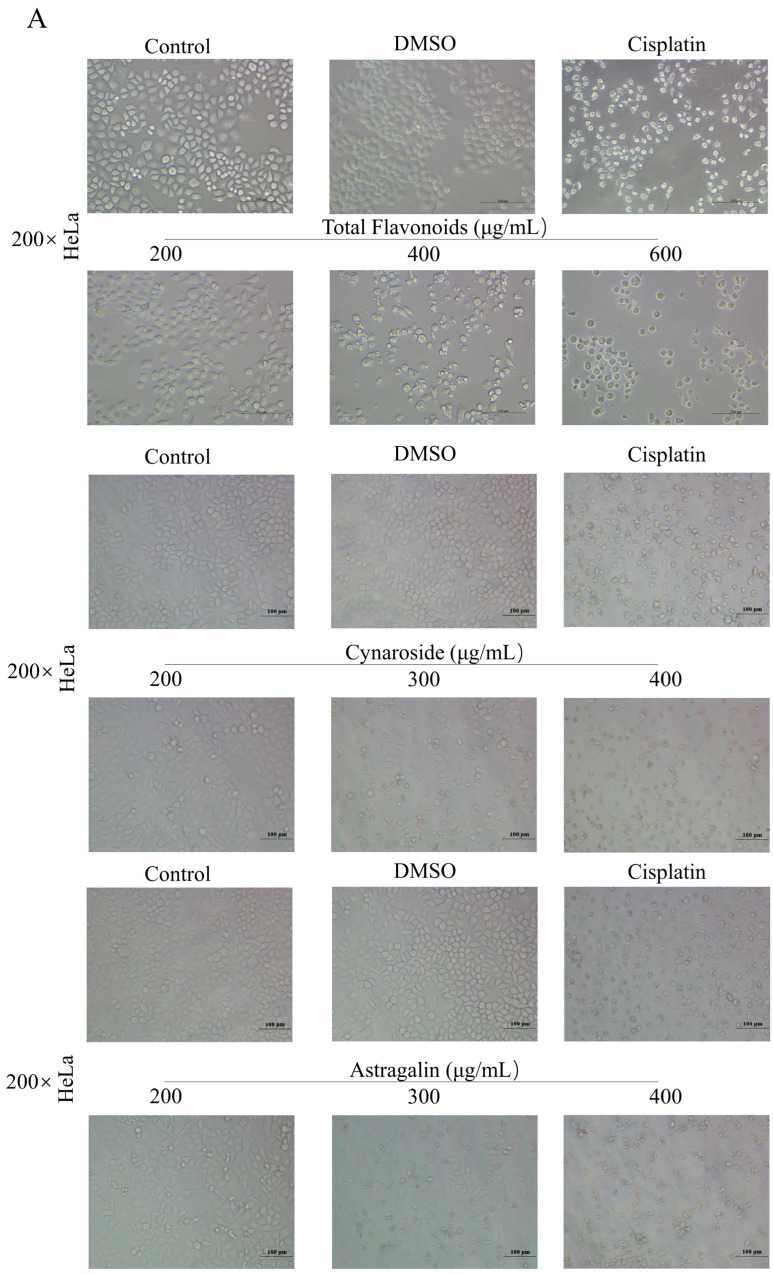
Total flavonoids and active ingredients can inhibit cell proliferation and induce apoptosis. Cells were treated with different concentrations of total flavonoids, Cynaroside, and Astragalin for 24 h, respectively. (**A**) The viability of HeLa cells was detected by MTT assay. (**B**) The morphology of HeLa cells was observed by inverted microscopy. (**C**) After staining with Annexin V and PI, HeLa cells were analyzed by flow cytometry. (**D**) The effect of Heochst 33342 staining on cell chromatin was observed under an inverted fluorescence microscope. ANOVA analyzed data. * *p* < 0.05, ** *p* < 0.01, *** *p* < 0.001, compared with control.

**Figure 9 ijms-24-16348-f009:**
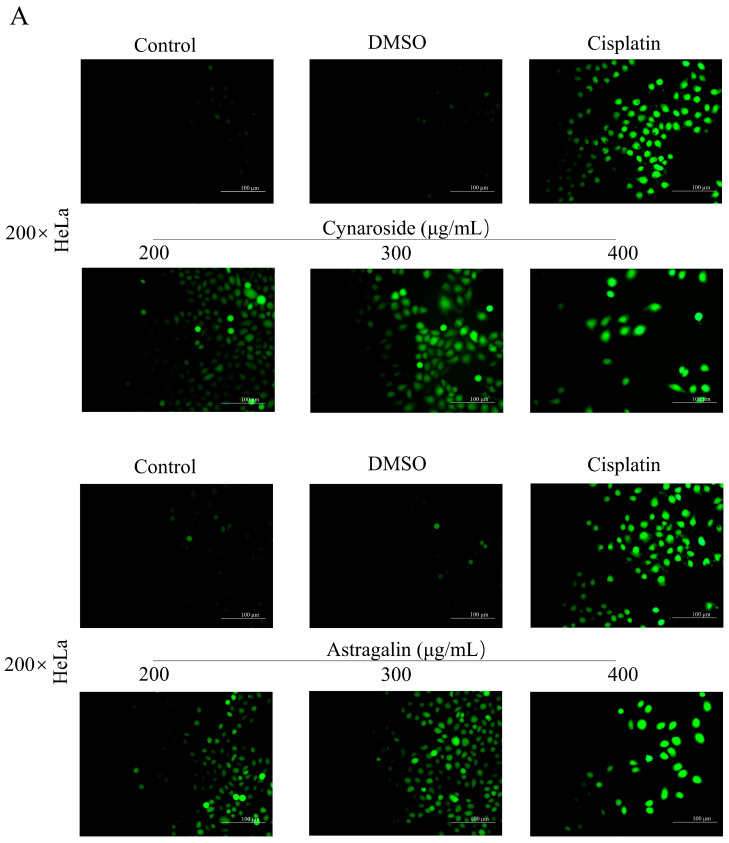
Induction of ROS accumulation by active ingredients of total flavonoids. Cells were treated with different concentrations of total flavonoids, Cynaroside, and Astragalin for 24 h, respectively. (**A**) Fluorescence intensity of ROS was observed by inverted fluorescence microscopy; (**B**) cellular ROS levels were detected by flow cytometry. ANOVA analyzed data. ** *p* < 0.01, *** *p* < 0.001, compared with control.

**Table 1 ijms-24-16348-t001:** Response surface experimental design for three factors: solid–liquid ratio, enzymatic hydrolysis temperature and ethanol concentration.

	A	B	C
Level	Solid-to-Liquid Ratio (g/mL)	Enzymatic Hydrolysis Temperature (°C)	Ethanol Concentration (%)
−1	12	40	70
0	15	45	85
1	18	50	100

**Table 2 ijms-24-16348-t002:** Response surface design and response results.

Tests	A-Solid-to-Liquid Ratio (g/mL)	B-Enzymatic Hydrolysis Temperature (°C)	C-Ethanol Concentration (%)	Extraction Yield (%)
1	−1	−1	0	2.4968
2	1	1	0	2.29127
3	−1	0	−1	2.72895
4	1	0	1	2.84924
5	0	0	0	3.62145
6	1	−1	0	2.51907
7	−1	0	1	2.85513
8	0	1	1	2.86716
9	0	−1	−1	2.8646
10	0	0	0	3.99514
11	0	0	0	3.69056
12	0	0	0	3.91323
13	1	0	−1	2.27592
14	−1	1	0	3.36115
15	0	0	0	3.83645
16	0	1	−1	2.61121
17	0	−1	1	3.22293

**Table 3 ijms-24-16348-t003:** Significance test for each regression coefficient and variance analysis.

Sources of Variation	Sum of Squares	Degree of Freedom (DOF)	Mean Square	*F* Value	*p* Value	Significance
Model	4.99	9	0.55	11.63	0.0019	significant
A	0.28	1	0.28	5.95	0.0448	
B	9.38 × 10^−5^	1	9.38 × 10^−5^	1.97 × 10^−3^	0.9659	
C	0.22	1	0.22	4.53	0.0709	
AB	0.3	1	0.3	6.26	0.0409	
AC	0.05	1	0.05	1.05	0.3399	
BC	2.62 × 10^−3^	1	2.62 × 10^−3^	0.055	0.8213	
A^2^	1.94	1	1.94	40.76	0.0004	
B^2^	0.91	1	0.91	19.11	0.0033	
C^2^	0.87	1	0.87	18.27	0.0037	
Residual	0.33	7	0.048			
Lack of Fit	0.24	3	0.079	3.33	0.1379	not significant
Pure Error	0.095	4	0.024			
Cor Total	5.32	16				
R^2^	0.9373					
CV%	7.14					

Note: *p* < 0.05 indicates a significant difference, and *p* < 0.01 indicates a highly significant difference.

**Table 4 ijms-24-16348-t004:** Compounds are contained in total flavonoids and MS parameters.

Number	Compound Name	Structured	Compound Class	Active	Tumor Type	Mechanism	References
**1**	Galangin	C_15_H_10_O_5_	Flavonoids	Anticancer	Ovarian cancer, colon cancer, liver cancer, gastric cancer, breast cancer,	Mitochondrial pathway induces apoptosis and cell cycle arrest and inhibits angiogenesis	[56,57,58,59,60]
**2**	Sinensetin	C_20_H_20_O_7_	Flavonoids	Anticancer	Gastric cancer, liver cancer, breast cancer	Regulation of AMPK/mTOR pathway	[61,62,63]
**3**	Orientin	C_21_H_20_O_11_	Flavonoids	Anticancer	Breast cancer, colon cancer, bladder cancer, cervical cancer	Inhibition of cell cycle arrest, inhibition of migration, and invasion	[64,65,66,67]
**4**	Hydroxygenkwanin	C_16_H_12_O_6_	Flavonoids	Anticancer	Liver cancer, breast cancer, lung cancer	Induces apoptosis and inhibits cell proliferation	[68,69,70]
**5**	Glycitein	C_16_H_12_O_5_	Flavonoids	Anticancer	Gastric cancer, glioma, breast cancer	NF-κB/AP-1-dependent and non-dependent pathways inhibit cell invasion and cell cycle arrest	[71,72,73]
**6**	Acacetin	C_16_H_12_O_5_	Flavonoids	Anticancer	Breast cancer, liver cancer, prostate cancer, lung cancer	Inhibition of cell cycle arrest and cell migration	[74,75,76,77]
**7**	Tiliroside	C_30_H_26_O_13_	Flavonoids	Anticancer	Liver cancer, breast cancer	Inhibits cell migration and invasion	[78,79]
**8**	Narcissoside	C_28_H_32_O_16_	Flavonoids	Antidiabetic			[80]
**9**	Tectochrysin	C_16_H_12_O_4_	Flavonoids	Anticancer	Colon cancer, lung cancer, prostate cancer	The death receptor pathway induces apoptosis	[81,82,83]
**10**	Daidzin	C_21_H_20_O_9_	Flavonoids	Anticancer	Cervical cancer	JAK2/STAT3 inhibits cell growth and induces apoptosis	[84]
**11**	Chrysin	C_15_H_10_O_4_	Flavonoids	Anticancer	Prostate cancer, gastric cancer, bladder cancer, colorectal cancer	Downregulation of PI3K-AKT and ERK inhibits cell proliferation, cell	[85,86,87,88]
**12**	Liquiritin	C_21_H_22_O_9_	Flavonoids	Anti-inflammatory, anticancer	cervical cancer, gastric cancer, liver cancer	Upregulation of p53 and p21 induces apoptosis	[88,89,90,91,92]
**13**	Quercetin	C_15_H_10_O_7_	Flavonoids	Anticancer	Gastric cancer, breast cancer, ovarian cancer, prostate cancer, liver cancer	Inhibition of cell cycle arrest, induction of apoptosis	[93,94,95,96,97]
**14**	Pinocembrin	C_15_H_12_O_4_	Flavonoids	Anticancer	Colon cancer, melanoma, ovarian cancer, lung cancer	Inhibition of cell autophagy, proliferation, and migration	[98,99,100,101]
**15**	Genkwanin	C_16_H_12_O_5_	Flavonoids	Anticancer	Lung cancer	Inhibition of cell cycle blockade	[102]
**16**	Isoliquiritigenin	C_15_H_12_O_4_	Flavonoids	Anticancer	Prostate cancer, breast cancer, lung cancer, cervical cancer	Inhibition of cell cycle blockade	[103,104,105,106]
**17**	Chalconaringenin	C_20_H_14_O_5_	Terpenoids	Antioxidant			[107]
**18**	Homoorientin	C_21_H_20_O_11_	Flavonoids	Anticancer	Colon cancer	Inhibition of cell proliferation and induction of apoptosis	[108]
**19**	Vitexin	C_21_H_20_O_10_	Flavonoids	Anticancer	Lung cancer, ovarian cancer, gastric cancer	Downregulation of p-p38/p38 and p-ERK1/2/ERK1/2; Bcl-2/Bax ratios	[109,110,111]
**20**	Eriodictyol	C_15_H_12_O_6_	Flavonoids	Anticancer	Lung cancer, cervical cancer	Inhibition of mTOR/PI3K/AKT signaling pathway and cell cycle blockade	[112,113]
**21**	Genistein	C_15_H_10_O_5_	Flavonoids	Anticancer	Ovarian cancer, breast cancer, cervical cancer	Inhibition of cell cycle arrest, induction of apoptosis	[114,115,116]
**22**	Tectorigenin	C_16_H_12_O_6_	Flavonoids	Anticancer	Ovarian cancer, breast cancer, liver cancer	Inactivation of AKT/IKK/IκB/NF-κB signaling pathway induces apoptosis	[117,118,119]
**23**	Rhoifolin	C_27_H_30_O_14_	Flavonoids	Anticancer	Breast cancer	Suppression of EMT	[120]
**24**	Apigenin	C_15_H_10_O_5_	Flavonoids	Anticancer	Pancreatic cancer, lung cancer, cervical cancer	Induction of cell cycle arrest and apoptosis	[121,122,123]
**25**	Schaftoside	C_26_H_28_O_14_	Flavonoids	Anticancer	Melanoma	Increased cellular autophagy	[124]
**26**	Diosmin	C_28_H_32_O_15_	Flavonoids	Anticancer	Breast cancer, colorectal cancer	Inhibition of cell cycle arrest, induction of apoptosis	[125,126]
**27**	Kaempferol	C_15_H_10_O_6_	Flavonoids	Anticancer	Ovarian cancer, colon cancer, breast cancer	Downregulation of PI3K/AKT and hTERT induces apoptosis and senescence	[127,128,129]
**28**	Isoquercitrin	C_21_H_20_O_12_	Flavonoids	Anticancer	Colon cancer, liver cancer	Downregulation of the PI3K-AKT/mTOR signaling pathway inhibits cell proliferation	[130,131]
**29**	Luteolin	C_15_H_10_O_6_	Flavonoids	Anticancer	Cervical cancer, breast cancer, colorectal cancer	Cell cycle arrest, inhibition of cell growth, and invasion	[132,133,134]
**30**	Hyperoside	C_21_H_20_O_12_	Flavonoids	Anticancer	breast cancer, Lung cancer	AKT/mTOR-p70S6K signaling pathway induces autophagy	[135,136]
**31**	Isorhamnetin	C_16_H_12_O_7_	Flavonoids	Anticancer	Lung cancer, colorectal cancer	Inhibition of PI3K AKT/mTOR signaling pathway to sup-press cell prolifera-tion	[137,138]
**32**	Diosmetin	C_16_H_12_O_6_	Flavonoids	Anticancer	Liver cancer	Inhibits cell cycle arrest and induces apoptosis	[139]
**33**	Protocatechualdeh-yde	C_7_H_6_O_3_	Phenols	Anticancer	Breast cancer	Cell cycle arrest	[140]
**34**	Kaempferol-3-O-rutinoside	C_27_H_30_O_15_	Flavonoids	Anti-inflammatory	Lung cancer	Inhibition of cancer cell growth through calcium signaling pathway calcium signaling pathway	[141]
**35**	protocatechuic acid	C_7_H_6_O_4_	Phenols	Anticancer	Lung cancer	Induction of apoptosis	[142]
**36**	Rutin	C_27_H_30_O_16_	Flavonoids	Anticancer	Colon cancer, lung cancer	Induction of apoptosis	[143,144]
**37**	Ononin	C_22_H_22_O_9_	Flavonoids	Anticancer	Breast cancer	Promotes cell apoptosis; decreases cell invasion and migration	[145]
**38**	Cynaroside	C_21_H_20_O_11_	Flavonoids	Anticancer	Gastric cancer.	Cell cycle arrest	[146]
**39**	Astragalin	C_21_H_20_O_11_	Flavonoids	Anticancer	Lung cancer, gastric cancer	Inhibits cell proliferation, cell migration, and invasion	[147,148]
**40**	2-Chloro-L-phenylalanine	C_9_H_10_ClNO_2_	Amino acids and their derivatives	Antioxidant			[149]
**41**	Naringenin	C_15_H_12_O_5_	Flavonoids	Anti-inflammatory, anticancer	Lung cancer, breast cancer	Inhibition of cell proliferation, cell cycle arrest	[150,151,152,153]

## Data Availability

The datasets used and/or analyzed during the current study are available from the corresponding author on reasonable request.

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
