# Peer review of "Total Flavonoids in Artemisia absinthium L. and Evaluation of Its Anticancer Activity"

_ijms, 2023, doi:10.3390/ijms242216348_

Round 1

Reviewer 1 Report

Comments and Suggestions for Authors

Review of the article "Total Flavonoids in Artemisia absinthium L. and Evaluation of its Anticancer Activity" by He et al.

About the abstract: at the beginning, it should outline the objective of the work, then a description of the methods used, and then a description of the results obtained. Certainly, it should be improved.

About introduction: long, insightful, introducing the topic of research, supported by many (50) references.

About results: written very carefully, clearly explained reasons and conclusions resulting from research. I have a question: Figure 3B: compound 41 naringenin has exactly the configuration as in the presented structure, or is it (±)? In my opinion, the authors should include in Supplementary Materials all obtained standard curves, such as those from section 2.1.

About the discussion: specifically written, engaging in discussion with literature data

About materials and methods: no major remarks, well written, minor editing remarks: lines 521 and 527 "Artemisia absinthium" should be written italic, line 539 should be mL instead of ml.

About conclusions: presents the most important data obtained in the discussed article.

About references: most of the items are from the last 5 years, moreover, it is very extensive, with as many as 156 items

Author Response

Dear Editors and Reviewers:

We appreciate the positive comments about the manuscript. We have modified this article according to your requirements.

Reviewer(s)' Comments to Author:

Reviewer: 1
Comments to the Author
About the abstract: at the beginning, it should outline the objective of the work, then a description of the methods used, and then a description of the results obtained. Certainly, it should be improved.

R: Thank you for your positive suggestion. We have improved the abstract. The modifications are as follows:

Abstract: To overcome the shortcomings of traditional extraction methods, such as long extraction time and low efficiency, and considering the low content and high complexity of total flavonoids in Artemisia absinthium L., in this experiment, we adopted ultrasound-assisted enzymatic hydrolysis to improve the yield of total flavonoids, and combined with molecular docking and network pharmacology to predict its core constituent targets, so as to evaluate its antitumor activity. The content of total flavonoids in Artemisia absinthium L. reached 3.80±0.13%, and the main components included Astragalin, Cynaroside, Ononin, Rutin, Kaempferol-3-O-rutinoside, Diosmetin, Isorhamnetin, and Luteolin. Cynaroside and Astragalin exert their cervical cancer inhibitory functions by regulating several signaling proteins (e.g. EGFR, STAT3, CCND1, IGFIR, ESR1). Gene Ontology (GO) and Kyoto Encyclopedia of Genes and Genomes (KEGG) pathway enrichment analysis showed that the anti-cancer of both compounds was associated with ErbB signaling pathway and FoxO signaling pathway. MTT results showed that total flavonoids of Artemisia absinthium L. and its active components (Cynaroside and Astragalin) significantly inhibited the growth of HeLa cells in a concen-tration-dependent manner with IC50 of 396.0±54.2 μg/mL and 449.0±54.8 μg/mL, respectively. Furthermore, its active components can mediate apoptosis by inducing the accumulation of ROS.

About introduction: long, insightful, introducing the topic of research, supported by many (50) references.

R: Thank you for your positive suggestion.

About results: written very carefully, clearly explained reasons and conclusions resulting from research. I have a question: Figure 3B: compound 41 naringenin has exactly the configuration as in the presented structure, or is it (±)? In my opinion, the authors should include in Supplementary Materials all obtained standard curves, such as those from section 2.1.

R: Thank you for your positive suggestion. The naringenin we isolated is shown in the figure and is not a racemate (±).

Quercetin is a flavonoid compound found in many plants. In laboratory studies, standard curves are commonly made with different concentrations of quercetin so as to quantify the flavonoids in the samples. The standard curve is as follows:

We also searched some of the literature for quantifying total flavonoids in terms of quercetin equivalents.

  1. Barakat, H.; Alkhurayji, R. I.; Aljutaily, T., Immune-Boosting Potentiating Properties of Brassica nigra Hydroalcoholic Extract in Cyclophosphamide-Induced Immunosuppression in Rats. Foods 2023, 12, (19).
  2. Chengolova, Z.; Ivanov, Y.; Godjevargova, T., Comparison of Identification and Quantification of Polyphenolic Compounds in Skins and Seeds of Four Grape Varieties.Molecules 2023, 28, (10).

About the discussion: specifically written, engaging in discussion with literature data.

R: Thank you for your positive suggestion. 

About materials and methods: no major remarks, well written, minor editing remarks: lines 521 and 527 "Artemisia absinthium" should be written italic, line 539 should be mL instead of ml.

R: Thank you for your positive suggestion. We have changed the lines 521 and 527 "Artemisia absinthium" to italics and line 539 "ml" to "mL". The modifications are as follows:

The influence of other factors on the extraction rate of flavonoids from A. absinthium L. was investigated using a single-factor experimental design.

Three factors, including solid-liquid ratio, enzymatic hydrolysis temperature and eth-anol content, were selected to significantly affect the extraction rate of total flavonoids from A. absinthium L.

It was then transferred to 2 mL glass vials and stored at -80 °C until analysis by UHPLC-MS/MS.

About conclusions: presents the most important data obtained in the discussed article.

R: Thank you for your positive suggestion. 

About references: most of the items are from the last 5 years, moreover, it is very extensive, with as many as 156 items.

R: Thank you for your positive suggestion.

Reviewer 2 Report

Comments and Suggestions for Authors

In the paper entitled "Total Flavonoids in Artemisia absinthium L. and Evaluation of its Anticancer Activity", He and co-workers described the extraction of the mixture of flavonoids from natural raw material using the ultrasound-assisted enzymatic hydrolysis and evaluated their activity against HeLa cells. Although the obtained results are interesting and worth publishing, the paper needs to be corrected due to several issues:

1.      Abstract - GO and KEGG abbreviations should be explained or avoided in the Abstract section.

2.      The introduction section should be corrected as now it is a mixture of topics without any fluent explanation of their relationship. For example, there is no explanation for why the Authors have chosen flavonoids from Artemisia in their studies. Were there any proofs of their anticancer activity? If yes, please give some literature examples.

3.      Huge efforts were put into optimizing the ultrasound-assisted enzymatic hydrolysis to extract flavonoids from A. absinthium. However, some comparison to other extraction techniques from the literature regarding yields should be provided to justify the method used.

4.      Figures 5 and 6 are too small to evaluate their content. Please enlarge the figures for better quality.

Author Response

Dear Editors and Reviewers:

We appreciate the positive comments about the manuscript. We have modified this article according to your requirements.

Reviewer(s)' Comments to Author:

Reviewer: 2

Comments to the Author
Abstract - GO and KEGG abbreviations should be explained or avoided in the Abstract section.

R: Thank you for your positive suggestion. We have explained the GO and KEGG abbreviations in the Abstract section. The modifications are as follows:

Gene Ontology (GO) and Kyoto Encyclopedia of Genes and Genomes (KEGG) pathway enrichment analysis showed that the anti-cancer of both compounds was associated with ErbB signaling pathway and FoxO signaling pathway.

The introduction section should be corrected as now it is a mixture of topics without any fluent explanation of their relationship. For example, there is no explanation for why the Authors have chosen flavonoids from Artemisia in their studies. Were there any proofs of their anticancer activity? If yes, please give some literature examples.

R: Thank you for your positive suggestion. A large amount of data suggests that flavonoids from Artemisia have anti-cancer activity, some of the references are listed below:[1]

  1. Taleghani, A.; Emami, S. A.; Tayarani-Najaran, Z., Artemisia: a promising plant for the treatment of cancer. Bioorg Med Chem 2020, 28, (1), 115180.
  2. Koyuncu, I., Evaluation of anticancer, antioxidant activity and phenolic compounds of Artemisia absinthium  L. Extract. Cell Mol Biol (Noisy-le-grand)2018, 64, (3), 25-34.
  3. Lang, S. J.; Schmiech, M.; Hafner, S.; Paetz, C.; Werner, K.; El Gaafary, M.; Schmidt, C. Q.; Syrovets, T.; Simmet, T., Chrysosplenol d, a Flavonol from Artemisia annua, Induces ERK1/2-Mediated Apoptosis in Triple Negative Human Breast Cancer Cells. Int J Mol Sci2020, 21, (11).
  4. Wei, X.; Xia, L.; Ziyayiding, D.; Chen, Q.; Liu, R.; Xu, X.; Li, J., The Extracts of Artemisia absinthium L. Suppress the Growth of Hepatocellular Carcinoma Cells through Induction of Apoptosis via Endoplasmic Reticulum Stress and Mitochondrial-Dependent Pathway. Molecules2019, 24, (5).
  5. Shawi, A.; Rasul, A.; Khan, M.; Iqbal, F.; Tonghui, M., Eupatilin: A flavonoid compound isolated from the artemisia plant, induces apoptosis and G2/M phase cell cycle arrest in human melanoma A375 cells. J. Pharm. Pharmacol2011, 5, (5), 582-588.

Huge efforts were put into optimizing the ultrasound-assisted enzymatic hydrolysis to extract flavonoids from A. absinthium. However, some comparison to other extraction techniques from the literature regarding yields should be provided to justify the method used.

R: Thank you for your positive suggestion. Lines 426 to 434 stated that the conversion rate using ultrasound-assisted enzymatic hydrolysis almost doubles compared to the enzymatic hydrolysis method, with a significant increase in productivity. We also reviewed other literatures for comparison with the ultrasound-assisted enzymatic hydrolysis method we used.

In this literature (Cheng, Y.; Zhao, H.; Cui, L.; Hussain, H.; Nadolnik, L.; Zhang, Z.; Zhao, Y.; Qin, X.; Li, J.; Park, J. H.; Wang, D., Ultrasonic-assisted extraction of flavonoids from peanut leave and stem using deep eutectic solvents and its molecular mechanism. Food Chem 2024, 434, 137497.), the best extraction conditions were found to be at a 27% water content in DES/HO, for 43 min with 31:1 g/mL liquid/solid ratio, giving 2.980 mg/g of flavonoids through the response surface method.

In this literature (Ou, H.; Zuo, J.; Gregersen, H.; Liu, X.-Y., Combination of supercritical CO2 and ultrasound for flavonoids extraction from Cosmos sulphureus: Optimization, kinetics, characterization and antioxidant capacity. Food Chem 2024, 435, 137598.), the highest total flavonoids content achieved was 1.99 ± 0.02 g RE/100 g at 25 MPa pressure, 55 °C temperature, 10% cosolvent concentration, and 0.21 W/mL UED using RSM optimization.

Figures 5 and 6 are too small to evaluate their content. Please enlarge the figures for better quality.

R: Thank you for your positive suggestion. We have enlarged the figures for better quality.

Round 2

Reviewer 2 Report

Comments and Suggestions for Authors

Dear Authors,

Almost all my issues have been addressed. The only thing left is still the Introduction section. For me, it is still not well explained why you have chosen this herb. There is a lack of a hypothesis, which is essential for research papers. The statement "Many studies have reported that flavonoids have a significant role in treating and preventing tumours [28-31]." is not enough. You have to give a more detailed explanation of the advantage of flavonoids from A. absinthi in comparison to other ones.

Author Response

Dear Editors and Reviewers: We appreciate the positive comments about the manuscript. We have modified this article according to your requirements. Reviewer(s)' Comments to Author: Reviewer: 3 Comments to the Author Almost all my issues have been addressed. The only thing left is still the Introduction section. For me, it is still not well explained why you have chosen this herb. There is a lack of a hypothesis, which is essential for research papers. The statement "Many studies have reported that flavonoids have a significant role in treating and preventing tumours [28-31]." is not enough. You have to give a more detailed explanation of the advantage of flavonoids from A. absinthi in comparison to other ones. R: Thank you for your positive suggestion. Wei et al. [1] found that the extract of Artemisia absinthium inhibited the growth of hepatoma cells through the induction of apoptosis that might be mediated by endoplasmic reticulum stress and the mitochondria-dependent pathway, and that three different organic-phase extracts effectively inhibited the growth of H22 cells in vivo in a mouse model of H22 tumors, and improved the survival rate of tumor-bearing mice, without significant toxicity. Meanwhile, Nazeri et al. [2] found that the methanolic extract of Artemisia absinthium triggered the apoptotic process in human colorectal cancer HCT-116 cells via down-regulation of Bcl-2 and activation of caspase-3 and BAX. Shafi et al. [3] reported that Artemisia absinthium inhibits proliferation of human breast cancer MDA-MB-231 and MCF-7 cells through the induction of apoptosis by regulating Bcl-2 family proteins and MEK/MAPK signaling. Therefore, we chose this herb, Artemisia absinthium, for the study of its antitumor activity. Craciunescu et al. [4] showed that the total phenolic acid and total flavonoid contents of Artemisia absinthium L. extracts were significantly higher compared to Arnica montana L., and that in recent years, Artemisia absinthium L. has attracted great attention in cancer therapy due to its total phenolic and total flavonoid contents [2]. Total phenols and total flavonoids, which are important natural constituents of Artemisia absinthium L., showed significant inhibitory effects on Huh-7 cell line, but were not toxic to Vero cell line [5]. Thus, we wanted to explore the anti-tumor mechanism of flavonoids in Artemisia absinthium L.. 1. Wei, X.; Xia, L.; Ziyayiding, D.; Chen, Q.; Liu, R.; Xu, X.; Li, J., The Extracts of Artemisia absinthium L. Suppress the Growth of Hepatocellular Carcinoma Cells through Induction of Apoptosis via Endoplasmic Reticulum Stress and Mitochondrial-Dependent Pathway. Molecules 2019, 24, (5). 2. Nazeri, M.; Mirzaie-Asl, A.; Saidijam, M.; Moradi, M., Methanolic extract of Artemisia absinthium prompts apoptosis, enhancing expression of Bax/Bcl-2 ratio, cell cycle arrest, caspase-3 activation and mitochondrial membrane potential destruction in human colorectal cancer HCT-116 cells. Mol Biol Rep 2020, 47, (11), 8831-8840. 3. Shafi, G.; Hasan, T. N.; Syed, N. A.; Al-Hazzani, A. A.; Alshatwi, A. A.; Jyothi, A.; Munshi, A., Artemisia absinthium (AA): a novel potential complementary and alternative medicine for breast cancer. Mol Biol Rep 2012, 39, (7), 7373-7379. 4. Craciunescu, O.; Constantin, D.; Gaspar, A.; Toma, L.; Utoiu, E.; Moldovan, L., Evaluation of antioxidant and cytoprotective activities of Arnica montana L. and Artemisia absinthium L. ethanolic extracts. Chem Cent J 2012, 6, (1), 97. 5. Ali, M.; Iqbal, R.; Safdar, M.; Murtaza, S.; Mustafa, G.; Sajjad, M.; Bukhari, S. A.; Huma, T., Antioxidant and antibacterial activities of Artemisia absinthium and Citrus paradisi extracts repress viability of aggressive liver cancer cell line. Mol Biol Rep 2021, 48, 7703-7710.

Round 3

Reviewer 2 Report

Comments and Suggestions for Authors

All my comments have been addressed. The paper can be published in its present form.